

# A new threat to local marine biodiversity: filamentous mats proliferating at mesophotic depths off Rapa Nui

Javier Sellanes[1,2], Matthias Gorny[3], Germán Zapata-Hernández[1,2,4], Gonzalo Alvarez[5,6], Praxedes Muñoz[1] and Fadia Tala[1,6,7]

[1] Departamento de Biología Marina, Facultad de Ciencias del Mar, Universidad Católica del Norte, Coquimbo, Coquimbo, Chile
[2] Millennium Nucleus for Ecology and Sustainable Management of Oceanic Islands (ESMOI), Universidad Católica del Norte, Coquimbo, Coquimbo, Chile
[3] Oceana, Santiago, Chile
[4] Programa de Doctorado en Biología y Ecología Aplicada (BEA), Universidad Católica del Norte, Coquimbo, Coquimbo, Chile
[5] Departamento de Acuicultura, Universidad Católica del Norte, Coquimbo, Coquimbo, Chile
[6] Centro de Investigación y Desarrollo Tecnológico en Algas y otros Recursos Biológicos (CIDTA), Facultad de Ciencias del Mar, Universidad Católica del Norte, Coquimbo, Coquimbo, Chile
[7] Instituto Milenio en Socio-Ecología Costera (SECOS), Santiago, Chile

Corresponding author
Javier Sellanes, sellanes@ucn.cl

## ABSTRACT

Mesophotic and deeper habitats ($\sim$40 to 350 m in depth) around Rapa Nui (Easter Island) were investigated using a remotely operated vehicle. We observed extensive fields of filamentous cyanobacteria-like mats covering sandy substrates and mostly dead mesophotic *Leptoseris* spp. reefs. These mats covered up to 100% of the seafloor off Hanga Roa, the main village on the island, located on its western side. The highest mortality of corals was observed at depths between 70 and 95 m in this area. Healthy *Leptoseris* reefs were documented off the northern and southeastern sides of the island, which are also the least populated. A preliminary morphologic analysis of samples of the mats indicated that the assemblage is composed of at least four filamentous taxa, including two cyanobacteria (cf. *Lyngbya* sp. and *Pseudoanabaena* sp.), a brown alga (*Ectocarpus* sp.), and a green alga (*Cladophora* sp.). An ongoing eutrophication process is suggested as a potential driver of the proliferation of these filamentous mats off Hanga Roa village.

## INTRODUCTION

Mesophotic coral ecosystems are deep reef communities that typically occur at a depth range of 30 or 40 to over 150 m (*Baker et al., 2016*). They are formed mainly by coral taxa adapted to living in low-light conditions and often also include other structure-forming taxa, such as sponge and macroalgae species (*Baker et al., 2016*; *Slattery & Lesser, 2021*). These ecosystems are now recognized as ecologically distinct and independent from their shallower counterparts and contain a substantial diversity of unique biota that is still

unexplored in most parts of the world (*Rocha et al., 2018*). The lack of knowledge about these deep coral ecosystems is a consequence of the difficulty of accessing the depths at which they occur, as technical diving (*e.g.*, rebreather diving using trimix) or sophisticated submarine equipment (*e.g.*, remotely operated vehicles, autonomous drop-cams, or manned submersibles) are required to carry out research. Mesophotic coral ecosystems are vulnerable to a series of anthropogenic stressors, such as fishing, thermal stress, diseases, pollution, invasive species, the marine aquarium trade, oil and gas exploration, cables, and pipelines (*Andradi-Brown et al., 2016*).

Rapa Nui (Easter Island; 27°07′S, 109°22′W), which formed ∼0.8 Mya, is a remote island located at the westernmost end of the large chain of seamounts comprising the Salas y Gómez ridge, relatively close to the East Pacific Rise (*Rodrigo, Díaz & González-Fernández, 2014*). Located in the easternmost apex of the Polynesian triangle, it is recognized for the high overall endemism levels of its coastal marine fishes (∼22%; *Randall & Cea, 2010*) and invertebrate taxa (4% to 34%; see *Fernandez et al., 2014*). However, this unique marine biodiversity is severely threatened by several anthropogenic impacts, including overfishing (*Zylich et al., 2014*), plastic pollution (*Hidalgo-Ruz et al., 2021*), exacerbated tourism (*Figueroa & Rotarou, 2016*), coastal erosion and terrestrial runoff (*Mieth & Bork, 2005*), and potential pollution from the percolation of domestic sewage and landfill contaminants into aquifers (*Rosa, 2013*).

Recently (2015–2018), through the use of a remotely operated vehicle (ROV), we have been able to access unexplored marine habitats (from ∼40 to 350 m deep) around the island, as well as at nearby seamounts, allowing for a first assessment of the biodiversity of mesophotic ecosystems and deeper sites (*Easton et al., 2019*), generation of new records of fauna, including fishes (*e.g.*, *Easton et al., 2017*) and echinoderms (*Mecho et al., 2019*), and reports of vast fields of the solitary mesophotic mushroom coral *Cycloseris vaughani* (*Hoeksema, Sellanes & Easton, 2019*). In these surveys, a chance discovery was the presence of dense and extensive fields of filamentous mats, covering the seafloor and nearby reefs at mesophotic depths at several locations around the island. It is known that cyanobacteria are a common constituent of coral reef ecosystems (*Stal, 2000*) and play an important role in nitrogen fixation and primary production (*Charpy et al., 2012*). However, under certain conditions, they can undergo massive proliferation, affecting the health of the ecosystem (*Bakker et al., 2017*; *Ford et al., 2017*). These events have been associated with variation in irradiance, nutrient supply, and other natural and anthropogenic disturbances (*Ford et al., 2018*). These proliferation events seem to be increasing at a global scale because of alterations in local biogeochemical cycles related to climate change (*Paul, 2008*; *Paerl & Paul, 2012*). These filamentous mats could develop into such dense blooms that they could even wash ashore, producing a mass accumulation, as reported by *Nagle & Paul (1999)* for Guam. At this location, benthic marine cyanobacterial blooms often occur in the presence of diverse assemblages of herbivorous fishes and urchins, but the underlying factors causing these proliferations, as well as the interaction mechanisms between grazers and these mats (since cyanobacteria are known to produce feeding-deterrent compounds), are still poorly understood (*Cissell, Manning & McCoy, 2019*; *Ford et al., 2021*). In addition, cyanobacteria have been directly linked with ciguatera fish poisoning outbreaks (*Laurent et al., 2008*),

and mats can create suitable habitats for other toxic microalgae, including toxin-producing dinoflagellates, thus generating co-occurring blooms (*Paerl & Otten, 2013*). Although several microalgae species are not toxic, their growth could produce low oxygen conditions as a consequence of organic matter accumulation and associated degradation processes in the bottom water, thus affecting the benthic communities (*Albert et al., 2012*). It is also possible that the rise of fixed nitrogen may modify its budget in the system, promoting the growth of macroalgae, further increasing the organic matter content within the sediments, and decreasing porewater oxygen content (*Brocke et al., 2015*; *Brocke et al., 2018*). In some environments, mats form associations with sulfate-reducing bacteria, producing sulfide, which is toxic for corals and establishes black band disease (*Myers & Richardson, 2009*; *Charpy et al., 2012*).

It has also been reported that in littoral reefs, green algae (chlorophytes) are common indicators of eutrophication (*Barile, 2004*). Most of the species in this group proliferate due to increased nutrient inputs, tolerate a wide range of environmental conditions, aggressively compete against sensitive corals, and have sub-lethal effects on several of the biological functions of corals (*Koop et al., 2001*; *Fabricius, 2005*; *Birrel et al., 2008*).

In this context, the aims of the present study were: (1) to provide a first approach to the spatial coverage of filamentous mats in the benthic ecosystem around Rapa Nui, (2) to evaluate the extent of the mesophotic coral reefs potentially impacted by these mats, and (3) to provide a preliminary description of the taxonomic composition of these mats.

## MATERIALS AND METHODS

Rapa Nui is a triangular-shaped island, delimited by the volcanoes Rano Kau in the southwest, Terevaka in the north, and Poike in the east, with Hanga Roa, the main village, located on the western side (Fig. 1). Aiming to have a representative spatial and bathymetric (~40 to 350 m deep) characterization of the mesophotic habitats on the three sides of the island, a remotely operated vehicle (ROV), controlled from local fishing boats, was deployed in 56 mostly independent sites around the island. There were 18 deployments each in January 2018 and 2019, and 20 during November and December 2019 (Fig. 1). The ROV, model Commander MKII (Mariscope Meerestechnik, Kiel, Germany), was equipped with two laser pointers, 10 cm apart, and a front-pointing HD video camera (Panasonic SD 909), angled at 45° and recording at 30 fps with a resolution of 1920 × 1080 pixels. The videos were analyzed at half their normal speed using GOM Player 2.3.19 (GOM & Company; https://www.gomlab.com/).

As mentioned in the Introduction, some of the results of these and previous ROV surveys have been presented elsewhere (*Easton et al., 2017*; *Easton et al., 2019*; *Hoeksema, Sellanes & Easton, 2019*; *Mecho et al., 2019*), for selected biotic components. For the present study, however, the focus was to evaluate the spatial coverage of filamentous mats in the benthic ecosystem, and the extent of mesophotic reefs potentially impacted by these mats, as well as their overall health conditions. The presence and coverage of filamentous mats were assessed semi-quantitatively by observing the seafloor in a stepwise manner as the ROV advanced over the ground along transects. Bottom-time varied between 10 and 42 min

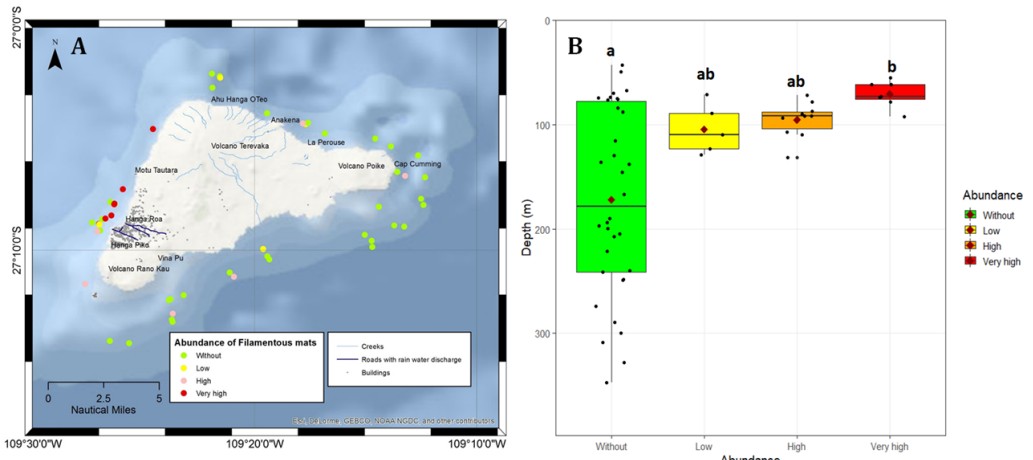

**Figure 1** **Map of Rapa Nui showing the main features of the island, the sites surveyed in the present study, and the extent of coverage of filamentous mats at these sites.** (A) Abundance of filamentous mats in the benthic ecosystems at the survey sites. (B) Depth range of the remotely operated vehicle stations for each category of filamentous mat coverage used in this study. Green: no mats observed, yellow: low coverage, orange: high coverage, and red: very high coverage. The box plots show the mean (red diamonds), median (horizontal black line), and lower and upper quartiles; the whiskers indicate the depth range for each category. The letters (a, b) indicate homogenous groups identified using the Wilcoxon *post hoc* test.

(mean: 25 min) per transect. In general, a portion of 10 to 20 min of video, considering mainly those segments in which the ROV was displaced at a steady velocity and a suitable distance from the bottom, was selected and analyzed per site. For each transect, we analyzed an area of at least 10 m$^2$, corresponding to ~15 non-overlapping frames. We exclusively analyzed those frames when the ROV was approximately 25 cm above the ground or in front of the reefs. As calibrated with the ROV on land, at these distances the images covered an area of ~0.65 m$^2$ (width ~117 cm × height ~65 cm).

According to the extent to which the bottom or the coral was covered by filamentous mats, the transects were cataloged into four groups: (1) without patches of filamentous mats, (2) low coverage (less than 50% coverage in at least five non-overlapping frames of the video of a transect), (3) high coverage (50% to 75% coverage in at least five frames of the video of a transect), and (4) very high coverage (100% coverage). Statistical comparisons of the mean depth between the four categories of filamentous mat coverage were evaluated using the Kruskal-Wallis test. *Post hoc* analyses were performed using pairwise comparisons with the Wilcoxon rank sum test and the *p*-value was adjusted using the Holm method (*Holm, 1979*). Before comparisons, normality and homogeneity of variance were tested using the Shapiro–Wilk and Levene tests, respectively. Statistical analyses were performed using RStudio (*R StudioTeam, 2020*), specifically the "car" package (*Fox & Weisberg, 2019*) and the "ggplot2" package for boxplots (*Wickham, 2016*).

We used the same ROV survey approximation to assess the extent of live coral coverage as a proxy for coral health status. Three categories were considered: (1) a healthy reef with >75% of the corals alive, (2) some damage with 25% to 75% of the corals alive, and (3) mostly damaged with <25% of the corals alive (mainly dead corals or fragments). Dead
corals were easily identified by their generally greenish or darker colors. Some were also covered by filaments.

To characterize the taxonomic composition of the filamentous mat assemblage, in May 2019, a small benthic trawl with a horizontal aperture of 30 cm was deployed at a site off Hanga Roa, where patches with 100% coverage were frequent. Mat samples were fixed using a 4% aqueous solution of formaldehyde (ACS Reagent; Sigma-Aldrich, St. Louis, MO, USA). For morphological characterization, filaments were observed using an Olympus IX71 inverted microscope equipped with phase contrast and epifluorescence (Olympus Co., Tokyo, Japan). Micrographs were taken using a camera ProgRes C3 (JENOPTIK AG, Jena, Germany), and measurements of cells (length and width) were carried out using ProgRes® CapturePro (JENOPTIK AG) analytical software. Monographic publications, floristic studies, and systematic articles were used for taxonomic identification of the macroalgae composing the mats, at least to the genus level (*Santelices, 1989*; *Loiseaux-de Goër & Noailles, 2008*; *Cormaci, Furnari & Alongi, 2014*; *Ramirez et al., 2018*). Guides and systematic articles were used to identify the cyanobacteria inhabiting the mat samples (*Komarek & Anagnostidis, 2007*; *Yu et al., 2015*; *Brocke et al., 2018*; *Zubia et al., 2019*). The identification of taxa was performed at the genus or species complex level.

Sample collection was performed with permission Res. Ext No. 41/2016 and No. 3314/2017 from SUBPESCA (National Fishing Authority of Chile) granted to the Universidad Católica del Norte. This project was also presented to the local *Consejo del Mar de Rapa Nui* (Council of the Sea of Rapa Nui), which permitted the capture of underwater footage and sampling around the island.

## RESULTS

The ROV transects around the island covered a depth range of 43 to 347 m. This allowed us to visualize the spatial and bathymetric distribution of sites with different levels of filamentous mat coverage (Figs. 1 and 2), and the distribution of mesophotic reefs and their health status around the island (Fig. 3). Mesophotic corals were represented by reef-forming *Porites lobata* and *Pocillopora* spp. at shallower depths (<60 m), *Leptoseris* spp., and *C. vaughani* at depths between 70 and 117 m, and sea-whips (*Stichopathes* spp.) between 127 and 327 m (Fig. 2). Other scleractinians were occasionally sighted deeper than 120 m (*e.g.*, cup corals), but they were too small to identify using ROV images.

### Spatial distribution of filamentous mats and corals

Filamentous mats were absent (category: without) from 34 of the studied sites around Rapa Nui, and low to very high coverage was observed at the remaining 22 sites, commonly on the western side of the island (Fig. 1A) and in water shallower than ~130 m (Fig. 1B). Statistical comparisons confirmed the significant differences between the depths of the mat-coverage categories (Kruskal-Wallis, Ch $i^2 = 12.9$, $df = 3$, $p = 0.005$), in particular between the categories without and very high (Wilcoxon test, $p = 0.023$; Fig. 1B). Other comparisons between categories were not significant (Wilcoxon test, $p > 0.05$). Indeed, high coverage was observed in the northwest corner (close to Hanga O'teo) at a depth of

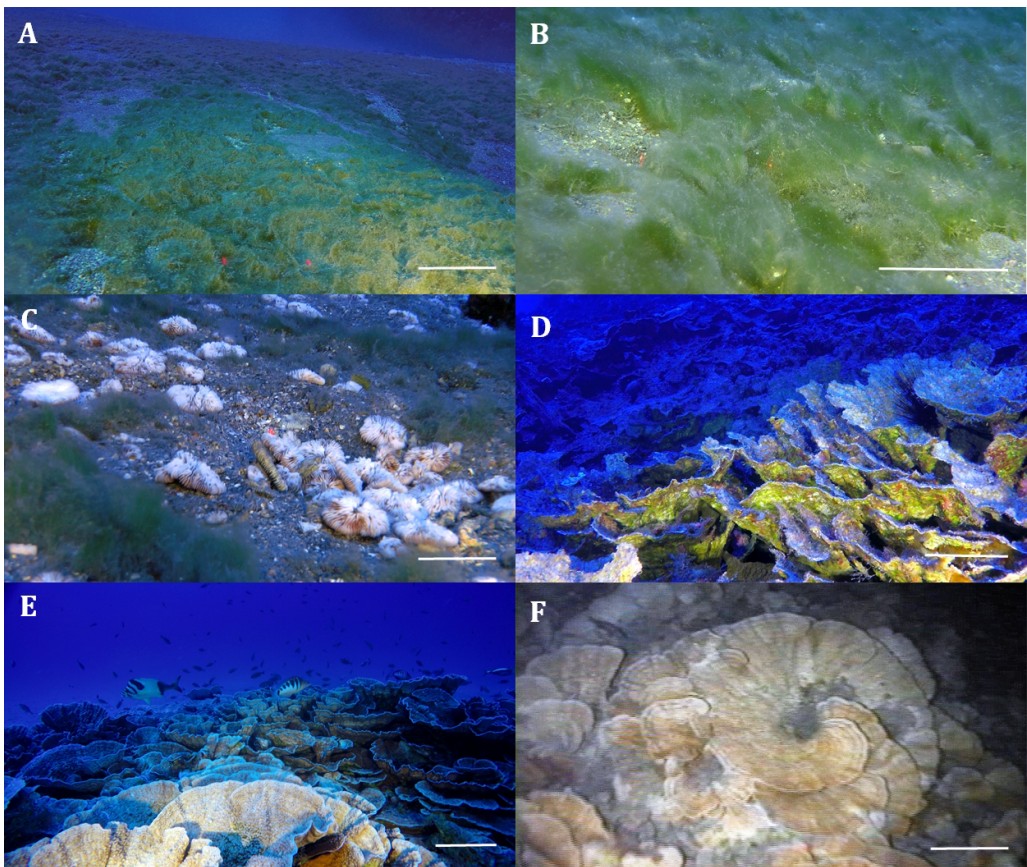

**Figure 2 Remotely operated vehicle (ROV) images of the filamentous mats and mesophotic reefs off Rapa Nui.** (A) Field of filamentous mats at ~80 m deep off Hanga Roa, Rapa Nui. (B) Close up view of the filaments. (C) Filaments among *Cycloseris vaughani* individuals. (D) Dead *Leptoseris* reef ~80 m deep overgrown by filaments. (E) Healthy *Leptoseris* reef off Anakena 80 m deep. (F) Healthy *Leptoseris* reef off Hanga Roa filmed during prospective ROV surveys during the "CIMAR-5 Islas" cruise conducted in 1999. Scale bars: 10 cm (A, B, C) and 25 cm (D, E, F). Images: Matthias Gorny, OCEANA.

123 m, and high and very high coverages were observed mainly off Hanga Roa (Fig. 1B) from 70 to 95 m deep.

Corals were observed in 50% of the 56 transects (Fig. 3A), and *Leptoseris* was present in 11 of them. Off Hanga Roa, the location where filamentous mats were most frequent, they were observed covering the sediments (Figs. 2A, 2B), fringing fields of the zooxanthellate mushroom coral *C. vaughani* (Figs. 2C, 3A; see also *Hoeksema, Sellanes & Easton, 2019*), and close by dead *Leptoseris* reefs (~80 m deep), which were also overgrown by filamentous mats (Fig. 2D). Healthy *Leptoseris* reefs were documented mainly off the northern and southeastern parts of the island (*e.g.*, near Anakena, La Perouse, and Vinapú) at depths of 68 to 82 m (Fig. 3B). Of the six locations with the healthiest *Leptoseris* reefs, four of them had no filamentous mats, or a sporadic presence of them, whereas at the three sites where the reefs were completely dead, filamentous mat coverage was high or very high (see also Supplemental Information).
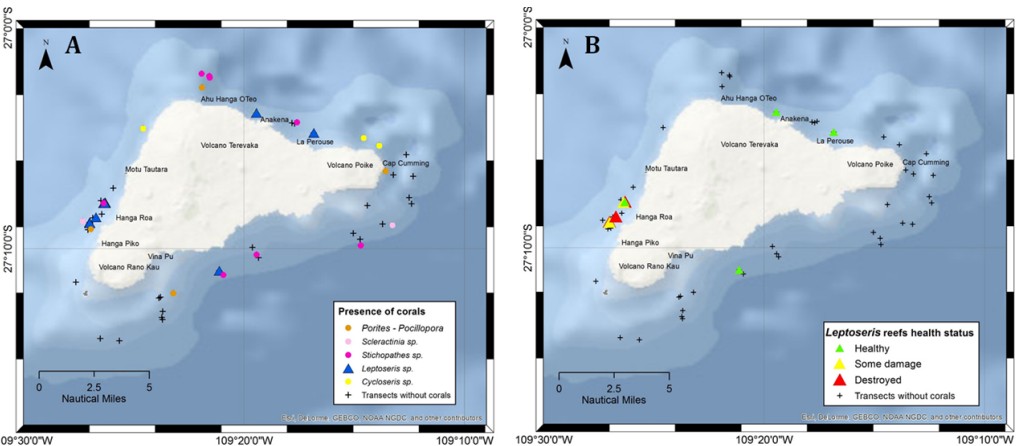

**Figure 3** **Transects surveyed off Rapa Nui in the present study showing sites with mesophotic corals.** (A) Map showing the main mesophotic coral taxa at each site. (B) Health status of *Leptoseris* reefs indicated by color: green = healthy (no noticeable impact), yellow = some damage (25%–75% of corals damaged), and red = destroyed (only dead corals or fragments observed).

## Taxonomic characterization of the filamentous mat assemblage

Morphological analyses of samples of mats collected off Hanga Roa indicated that mats are an assemblage of at least four taxa: one Chlorophyta (*Cladophora* sp.), one Ochrophyta (*Ectocarpus* sp.), and two Cyanobacteria (*Lyngbya* s.l. ([sensu lato] and *Pseudoanabaena* sp.) (Fig. 4) as follows:

*Cladophora* sp. (Figs. 4A, 4B): thallus of green to light green branched uniseriate filaments with 2–3 cm in total length. Basal part of the filaments fixed to the substrate by a primary rhizoid. Presence of unilateral branches inserted laterally or obliquely on the filament. Principal axis constituted by cylindrical cells measuring of 998.9 ± 69.2 μm in length and 223.3 ± 9.5 μm in diameter. Apical cells cylindrical, round ended with a diameter of 250.0 ± 7.6 μm and length of 701.8 ± 76.0 μm. Zoosporangia were not observed.

*Ectocarpus* sp. (Figs. 4C, 4D): thallus of light brown to olive sparingly branched filament 0.1–0.5 cm in total length. Cells conform to uniseriate filaments ending in a rounded apical cell. Cells barrel-shaped, 50.0 ± 7.3 μm in length, and 14.1 ± 3.4 μm in diameter. Plurilocular sporangia were present, elongated with cylindroconical form, 80–130 μm in length and 20–30 μm in diameter.

*Lyngbya* s.l. (Figs. 4E, 4F): thallus caespitose, brownish-red, filaments slightly curved, sheet colorless, lamellated with apices not attenuated at the end. Trichome not constricted at the cross-wall, cylindrical cells very short 3.5 ± 0.3 μm in length and 7.1 ± 0.1 μm in diameter, sheath 1.6 ± 0.3 μm, end cells rotund, calyptra absent.

*Pseudoanabaena* sp. (Figs. 4G, 4H): trichomes solitary or crowded in clusters, straight or almost straight, pale blue–green. Cells barrel-shaped, 2.8 ± 0.8 μm in length and 1.2 ± 0.1 μm in diameter, intensely constricted at cross walls, no heterocysts or sheath, end cells round.

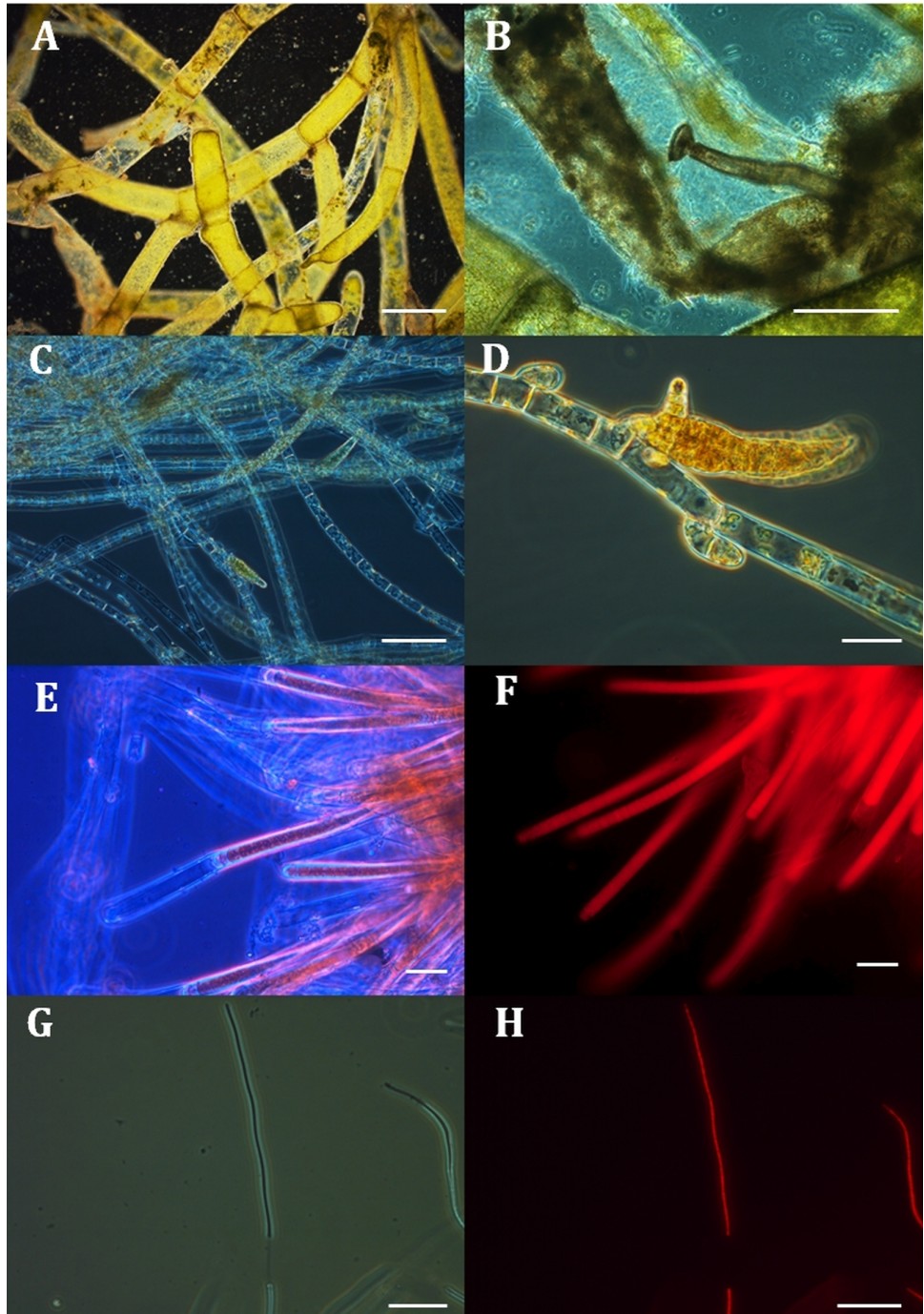

**Figure 4** **Micrographs of four filamentous taxa in samples from mats collected off Hanga Roa, Rapa Nui at mesophotic depths. A–E and G were photographed using phase-contrast and F and H using epi-fluorescence techniques.** (A, B) *Cladophora* sp. (C, D) *Ectocarpus* sp. (E, F) *Lyngbya* s.l. (G, H) *Pseudoanabaena* sp. Scale bars represent (A): 500 μm, (B): 200 μm, (C): 100 μm, (D): 30 μm, (E and F): 20 μm, and (G) and H: 30 μm.

## DISCUSSION

Although we have provided only a preliminary taxonomic characterization of the filamentous mats covering sandy areas and dead mesophotic reefs off Rapa Nui, our findings indicate that these mats are composed of at least two cyanobacteria. We are aware of, and recognize the limitations associated with our approach to identifying mat taxa, based only on morphology. As indicated by *Komárek (2016)*, *Lyngbya*, *Okeania*, and *Moorea* cannot be distinguished from each other using light microscopy. Thus, we refer to *Lyngbya* s.l. ([sensu lato] and suggest that genetic analysis is needed to clarify this classification. Cyanobacteria-dominated microbial mats are known to be typical components of coral reef systems and often undergo massive proliferation (*Stal, 2000*). These events have been associated with natural processes (*e.g.*, variation in irradiance), but mostly with anthropogenic disturbances that increase nutrient concentrations in the marine environment (*Ford et al., 2017*). The highest coverage of mats was observed mainly off Hanga Roa village, which has the highest concentration of the island's human population (7,750 inhabitants; http://www.ine.cl) and where most tourists engage in recreational activities. *Figueroa & Rotarou (2016)* reported ~20,000 visitors per year in the late 1990s, whereas ~150,000 were reported during 2019 (http://www.sernatur.cl), representing an approximately eight-fold increase over the last two decades. Factors such as overtourism, the absence of a wastewater collection and treatment system (most of the residences have cesspools and a minor proportion have septic tanks), and the unlined landfill (*Rosa, 2013*) potentially pose a great threat to the marine environment off Hanga Roa village, owing to the potential input of organic matter, nutrients, and contaminants. Pollutants can reach the sea by runoff or percolation to aquifers that eventually discharge into the sea. On Rapa Nui, submarine groundwater discharges are ubiquitous in intertidal environments around the island (*Brosnan, Becker & Lipo, 2018*), and could hypothetically also seep through deeper sediments (*Montgomery EL & Associates INC, 2011*), potentially conducting nutrients of anthropogenic origin directly to mesophotic habitats. Indeed, very low salinities (4.7–16.8 psu) have been measured in the overlying water of unperturbed sediment cores obtained off Hanga Roa where filamentous mats proliferate, further suggesting percolation of pollutants to aquifers in the area (P. Muñoz, unpublished data). A similar situation has been observed at the western flank of Hawai'i Island, where freshwater from onshore aquifers can flow through permeable fractured basalts, mix with seawater to form freshened groundwater, and seep into offshore (mesophotic) benthic areas (*Attias et al., 2020*). Furthermore, the observation of low salinity bottom water is concomitant with relatively high $NO_3-$ concentrations (1.87 and 3.03 μM), compared to two other sites where nitrate concentrations were undetectable in overlying waters with normal salinities (~35 psu) (P. Muñoz, unpublished data). Therefore, it is feasible that the benthic fluxes and submarine groundwater discharges could channel nutrients to mesophotic depths, enhancing algal and cyanobacterial growth, to the detriment of corals. A similar situation, albeit caused by groundwater nutrients derived from bird guano, was observed in the coral reefs of Heron Island (Great Barrier Reef, Australia; *McMahon & Santos, 2017*). In addition to the potential impacts of pollutants, the permanent coastal erosion around Rapa Nui and

terrestrial runoff during rainy seasons (May to October) could also increase nutrient inputs to the coastal environments, including ammonia, nitrate, and silicate, which are known to have negative consequences for corals (*D'Angelo & Wiedenman, 2014*). Furthermore, the volcanic origin of Rapa Nui, together with enhanced erosion could also increase the iron concentration in the marine ecosystem. Iron from shipwrecks has been found to directly drive cyanobacteria expansion in iron-limited reefs in the Pacific (*Kelly et al., 2012*; *Mangubhai & Obura, 2018*). Increased iron added to a decrease in the N:P ratio could even further stimulate the proliferation of cyanobacteria (*Ford et al., 2018*).

Regarding mesophotic reefs, two species of the genus *Leptoseris* have been reported for Rapa Nui, *L. scabra* and *L. solida*, both collected in 1999 off Hanga Roa at depths of 43 m and 80 to 100 m, respectively (*Glynn et al., 2007*). Given the depth of our observations as well as the plate-like structure of the colonies, as indicated by *Glynn et al. (2007)*, the damaged reef off Hanga Roa village was probably composed mainly of *L. solida*. A piece of evidence, also obtained in November 1999 during the first ROV survey ever done off Hanga Roa at ∼80 m deep, suggests that the same *Leptoseris* reef that is currently dead was healthy ∼20 years ago (Fig. 2D; *Gorny & Retamal, 2000*). In the present study, live *Leptoseris* reefs were documented mainly off the northern and southeastern sides of the island (*e.g.*, Anakena, La Perouse, and Vinapú).

Despite the circumstantial indication of the health status of the mesophotic reefs off Hanga Roa a few decades ago, the ecological impacts on the biodiversity and ecosystem functioning associated with anthropogenic causes are still unknown and need further investigation in the short term. Further research should address a more detailed taxonomic characterization of these mats, for example, through molecular techniques; assessment of the seasonal, spatial, and structural patterns of the assemblage; their eventual role in reef deterioration; recognition of eutrophication mechanisms; and long-term monitoring of dissolved organic matter and nutrient dynamics. These studies are encouraged to inform the implementation of effective and integrated land-sea management actions, including a wastewater treatment system. This information should also be key to inform the implementation of management strategies of the recently created Marine Protected Area of Multiple Uses (MPA-MU) of Rapa Nui, currently the largest in Latin America. This protected area encompasses ∼579,000 km$^2$ (*Paredes et al., 2019*) and aims to protect this unique world biodiversity heritage site. In addition, this study will also serve as a baseline for future studies of changes in the mesophotic ecosystem off Rapa Nui after closure of the island to tourism, from March 2020 to date, due to the COVID-19 pandemic.

## CONCLUSIONS

Based on opportunistic video observations, we provide the first report of filamentous mats covering sandy areas and dead mesophotic reefs (*Leptoseris* spp.) off Rapa Nui. A preliminary morphological analysis of mat samples suggested that the assemblage is constituted by at least four filamentous taxa, including two cyanobacteria (*Lyngbya* s.l. and *Pseudoanabaena* sp.), a brown alga (*Ectocarpus* sp.), and a green alga (*Cladophora* sp.). Whereas a highly damaged, even completely dead, *Leptoseris* reef was observed in

the waters off the main village on the western side of the island, reefs in much healthier conditions were observed off the less populated northern and southeastern parts of the island (*e.g.*, Anakena, La Perouse, and Vinapú). Circumstantial evidence indicates that the *Leptoseris* reef off Hanga Roa was alive ~20 years ago. Our preliminary evidence suggests a link between ongoing eutrophication associated with human population expansion and deficient management of wastewater and urban runoff on the western side of the island, the proliferation of filamentous mats, and consequent damage to mesophotic *Leptoseris* reefs.

## ACKNOWLEDGEMENTS

We thank Poky Tane Haoa, Ricardo Hito, and Enrique Hey from the Rapa Nui community. We also thank Erin Easton and Ariadna Mecho for their collaboration during fieldwork, and Maria Valladares and Valentina Hevia for helping with the collection of the samples. Our thanks go to Bert Hoeksema for his valuable comments on a very early version of this manuscript, as well as to Amanda Ford, Marc Slattery, and an anonymous reviewer.

### Funding

This work was supported by grants FONDECYT 1180694 and 1181153, as well as funding from the ANID- Millennium Science Initiative ESMOI. Germán Zapata-Hernández was funded by CONICYT-PCHA/ Doctorado Nacional/2015-21151249 and Beca de Postdoctorado MINEDUC UCN-19101 N° 002. Oceana Chile contributed to the funding of the expeditions and provided the ROV. The funders had no role in study design, data collection and analysis, decision to publish, or preparation of the manuscript.

### Grant Disclosures

The following grant information was disclosed by the authors:
FONDECYT: 1180694, 1181153.
ANID- Millennium Science Initiative ESMOI.
CONICYT-PCHA/ Doctorado Nacional: 2015-21151249.
Beca de Postdoctorado MINEDUC: UCN-19101 No 002.
Oceana Chile.

### Competing Interests

The authors declare there are no competing interests.

### Author Contributions

- Javier Sellanes and Matthias Gorny conceived and designed the experiments, performed the experiments, analyzed the data, prepared figures and/or tables, authored or reviewed drafts of the paper, and approved the final draft.
- Germán Zapata-Hernández and Gonzalo Alvarez performed the experiments, analyzed the data, prepared figures and/or tables, authored or reviewed drafts of the paper, and approved the final draft.

- Praxedes Muñoz conceived and designed the experiments, performed the experiments, analyzed the data, authored or reviewed drafts of the paper, and approved the final draft.
- Fadia Tala performed the experiments, analyzed the data, authored or reviewed drafts of the paper, and approved the final draft.

### Field Study Permissions

The following information was supplied relating to field study approvals (i.e., approving body and any reference numbers):

Sample collection was performed with permission Res. Ext N° 41/2016 and N° 3314/2017 from SUBPESCA (National Fishing Authority of Chile) granted to the Universidad Católica del Norte. This project was also presented to the local Consejo del Mar de Rapa Nui (Council of the Sea of Rapa Nui), which permitted the capture of underwater footage and sampling around the island.

### Data Availability

The raw data are available in the Supplementary File.

### Supplemental Information

Supplemental information for this article can be found online at http://dx.doi.org/10.7717/peerj.12052#supplemental-information.

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
