# Peer review of "A new threat to local marine biodiversity: filamentous mats proliferating at mesophotic depths off Rapa Nui"

_PeerJ, doi:10.7717/peerj.12052_

## Round 0.1 · original submission · Major Revisions

Dear Javier and co-authors,

I have received three independent reviews of your study. While all reviewers clearly recognised the novelty of your work, they have collectively raised a number of major issues that will need to be addressed in your revised manuscript. In particular, all three reviewers
have highlighted that the current manuscript is of highly descriptive nature and that not enough data (and data analyses) are provided to support the conclusions. Two reviewers strongly suggested that you incorporate additional data and statistical analyses from the high-quality videos you have generated. It was also noted that (1) the revised manuscript should be carefully edited for grammatical structure by a fluent English writer, and (2) that the reference list is currently missing some relevant literature.

Overall, the reviewers have provided you with excellent suggestions on how to improve the manuscript, and I will be looking forward to receiving your revised manuscript along with a point-by-point response to the reviewers' comments.

With warm regards,
Xavier

·

Basic reporting

The English was of a quality that allowed this reviewer to understand the system/problem/data. However, there were some minor grammar issues and the revision would benefit from a review by a native English writer. PeerJ requirements are met throughout. My only basic concern is that the Introduction suffers from some misrepresentations of mesophotic reefs: 1) I would not characterize MCEs as “formed mainly by light-dependent corals” [line 42], rather they represent low-light adapted species (= a few coral species, but also sponges & gorgonians, etc. see for reviews: Lesser et al. [2018] Ann Rev Ecol Evol Syst 49:49-71; Slattery & Lesser [2021] Front Mar Sci 8:654268), 2) while it is true that these are vulnerable systems (Andradi-Brown et al. 2016), it is also worth noting that they may represent a refuge for depth generalists (Bongaerts et al. [2010] Coral Reefs 29:309-327; Slattery et al. [2011] JEMBE 32-41).

Experimental design

The experimental design utilized extensive ROV surveys from around the island to characterize mat and coral cover/health. Unfortunately, with virtually no data analysis, the paper provides an observational report and speculation regarding the drivers. I would urge the co-authors to include: 1) percent cover live vs dead corals, 2) numbers of fish (and species) at each site, 3) correlations with mat cover/sites around island. These quantitative data will be much more useful for the scientific community (ie, MCE vulnerability) and for conservation (ie, return sampling post-COVID), 4) can you include any nutrient data (collections or a meta-analysis of existing datasets).

Validity of the findings

The videos are very impressive and high quality; a little more work (as suggested above) will strengthen this manuscript and the conclusions. Statistic are necessary in these systems (note the response to cited: Rocha et al. [2018] Science361:281-284, Roberts et al. [2018] “The challenge of the deep should not preclude robust ecological analysis”. Science. sciencemag.org/content/361/6399/281/tab-e-letters).

Additional comments

This paper has the potential to be a significant contribution to the field.

·

Basic reporting

The paper by Sellanes et al. is overall a well-written and insightful piece of work that uses a good amount of appropriate references in their background information and interpretation. The manuscript structure and figures are clear and the raw data is shared in the appendix.

I would recommend the raw data to also include the information on coral cover that was collected, and to show the deviation/error of the depth, or perhaps minimum/maximum, as well as the mean. In terms of structure, my only comment is that the conclusion is currently very long, and I suggest the part of future research and its importance should be moved to the end of the discussion.

Experimental design

The research question addressed by Sellanes et al. is important and timely. There has recently been an increase in attention on benthic cyanobacteria and very little is known about their occurrence on mesophotic reef systems (due to the difficulties highlighted by the authors in studying this system). As such, the authors identify a clear research gap and define three clear aims (not research questions) linked to this topic. I do however have some reservations with the experimental design that I hope the authors can address in a revision.

In terms of the methods, I am struggling to understand the survey design in terms of depth at each site which has made it difficult to interpret the findings. The raw data, including coordinates and mean depth of each site, is provided in the appendix which is very helpful, but I suggest it should be clearer in the manuscript itself. The raw data suggests that there were in fact only 24 individual stations (as opposed to 56 as described in the methods) which were measured 1-3 times over 2018 and 2019. While it seems the coordinates of the stations remain similar, the depths change dramatically over the years (i.e. station 2 in 2018 was 200m, but in 2019 was 89.5m). If the stations are monitored more than once, the results need to change to account for this - on the other hand if these should all be considered separate stations then the authors should change the 'Station' numbers in the raw data table.

Depth would be expected to have an impact on benthic community structure (and the authors show this in Figure 1B), yet it remains unclear how the depths of the ROV surveys were incorporated into the study design in a standardised way. For example, are there any deep sites (>150m - where seemingly cyanobacteria are absent as per Figure 1B) on the western side where the authors postulate pollution is driving mat proliferation? This is also important to consider when comparing to the 1999 ROV survey (lines 216-218) to show that coral cover has declined. I would suggest the authors need to spend some time clarifying this throughout the respective sections (perhaps including a map that shows this).

Lastly, I struggle a bit to understand why the authors are basing their semi-quantitative interpretations on only 5 non-overlapping frames when they have 10-20 minutes of video data. How much area does each 'frame' cover? The selected approach provides only semi-quantitative information from a very small area, while there are ways to extract benthic community information from video transects that could provide much more detailed and accurate information. I think it is important the authors justify/explain this approach better in the text and that this needs to be more convincing, or otherwise more data added, prior to publication.

Validity of the findings

Overall the results are reported well, though in places there is certainly room for expansion. The validity of the findings and the authors' interpretations can be strengthened by incorporating more information on site depth (as suggested in an earlier point).

In terms of the taxonomic information, I recommend the authors to present their morphological ID of cyanobacterial taxa with caution. Perhaps the selected approach is much more powerful that bright field microscopy, but with bright field microscopy Lyngbya, Okeania, or Moorea cannot be distinguished from each other (see Duperron et al. 2020) and should thus be referred to as Lyngbya s.l. ([sensu lato], in a broad sense, as opposed to Lyngbya s.s. [sensu stricto], in a narrow sense) (as in Ford et al. 2021). I understand these to only be able to be differentiated by sequencing. Both mentioned references are included in my comments on the PDF, as well as others I recommend the authors to look into.

In terms of coral decline, the authors draw some large conclusions of 'catastrophic loss' but the current manuscript does not have the data to support this. In the absence of previous data, this could be done potentially by showing the amount of dead coral per site - coral loss from 90 to 0% is definitely catastrophic, while from 10 to 0% is less so as the system was never coral-dominated. Among other quantitative/statistical tests the authors could have incorporated to strengthen their interpretations, the authors miss an opportunity to test for correlation between mat abundance and Leptoseris reef health status, though they describe this relationship in the discussion. I recommend the authors to consider including some tests to make their interpretations of their data more robust.

Additional comments

Overall I enjoyed reading this paper and the findings related to cyanobacterial abundance on Easter Island's mesophotic reefs. I have added some specific comments into the PDF document that I hope will help the revision of this manuscript.

Reviewer 3 ·

Basic reporting

It is my belief that the manuscript will benefit from revisions of the writing style and use of english. Wording is at times unclear and gramatical as well as spelling mistakes are occasionally present. I list some examples and suggestions below.
* * *
Abstract

Lines 28-30
We have sighted vast fields of filamentous cyanobacteria-like mats, covering sandy bottoms, as well as over dead corals of mesophotic reefs (mainly composed by Leptoseris sp.).

I suggest this could be re-written as follows to improve clarity and also quation if the authors refer to a single species of Leptoseries observed or if different species of the genera were observed in which case spp. should be used instead of sp.

We observed extensive fields of filamentous cyanobacteria-like mats covering sandy substrates and dead corals on mesophotic reefs dominated by Leptoseris spp.


lines 30- 32: “(up to ~100% of total seafloor surface)”
I suggest minor text change as brackets are not the most appropriate way to report this result which can be placed in the sentence and that approximately 100 % is not the most appropriate way to refer the coverage of nearly all of the substrate or at times all of the substrate. It may also be useful to specify the level of coral mortality here as a percentage. I also suggest relying less on parenthesis throughout the manuscript and specifying results of information in the text. A suggestion is as follows:

“These mats cover up to 100% of the seafloor and also high mortality
of corals occurs between 70 and 95 m depth off Hanga Roa, the main village on the western side of the island.”
* * *
Introduction:

Lines 41 to 43:
I suggest revising
“Mesophotic coral reef ecosystems typically distribute from 30-40 m to over 150 m depth, they occur in tropical and subtropical waters and are formed mainly by light-dependent corals, but also including other typical reef organisms such as sponge and macroalgae species (Baker et al., 2016).”

To

“Mesophotic coral reef ecosystems are typically distribute from 30-40 m to over 150 m depth, occur in tropical and subtropical waters and are formed mainly by light-dependent corals, but also including other typical reef organisms such as sponge and macroalgae species (Baker et al., 2016).”


Line 58
I suggest the terminology “anthopogenic issues” is inconsistent with the text above which refers to “anthropogenic stressors” at line 49.

Line 82
Typo in referencing - no space between “Nagle” and “&” in “(Nagle& Paul, 1999,…”

Line 82 and line 91
I suggest revising the English usage of “also” in the following sentences:

“Cyanobacteria have been also directly linked…”
To
“Cyanobacteria have also been directly linked

“It has been also…”
to
“It has also been…”

Lines 91 and 92
I suggest the usage of plural and singular is confusing in:
“… green algae (i.e., Chlorophytes) are one of the most
common indicator species of eutrophication…”
This could be revised to “… green algae (i.e., Chlorophytes) are
common indicators of eutrophication…”

English usage is incorrect and confusing on lines 94 -96
“…constituting an aggressive competitive species against sensible corals, and having sub-lethal effects over several biological functions of them…”

I suggest revising in a way similar to:

“Most of the species in this group proliferate due to increased nutrient inputs and tolerate a wide range of environmental conditions, aggressively competing with corals, and have sub-lethal effects over several biological functions of corals”
* * *
Results
Lines 146 to 147
It is unclear to me if a single species from each of the genera Pocillopora, Leptoseris and Cycloseris were observed repeatedly. If it is likely that more than one species was observed then spp. should be used instead of sp.
* * *
Discussion
Line 190
Confusing wording and spelling issues in : “…suggested that they are compossed of at least to cyanobacteria”

"compossed" should be spelt "composed"
"to" should be "two"

Line 191
I suggest deleting the word inhabitants in “…microbial mats are typical inhabitants of coral reef systems…”

Lines 193-194
I suggest revising
“…with anthropogenic disturbances that generate an increase in the nutrient supply to the marine environment…” to
“…with anthropogenic disturbances that increase nutrients in the marine environment…”

Line 212
It is unclear if the term “damage” is used to refer to “mortality” of corals


Lines 218 to 219
Wording is confusing in:
“At that opportunity, the tourist number reached ~20.000 visitants per year (Figueroa & Rotarou, 2016).

I also suggest discussing this in the context of current numbers of tourists and clarifying how this is a proxy for the intensity of anthropogenic stressors. For example has the number of coastal residences increased and has the volume of sewage discharge increased?
* * *
Conclusion
Lines 228 -229
I suggest revising the sentance
“Catastrophic damage was observed at mesophotic Leptoseris sp. reefs off the main village of the island.”
To
“High mortality of Leptoseris sp. was observed on mesophotic reefs off the main village of the island.”

Experimental design

It is unclear from the manuscript if there is a specific design to the surveys. Rather I gain the impression that 56 opportunistic surveys of locations were undertaken as a result of vessels facilitating the surveys. The study therefore undertakes initial descriptions of mesophotic reefs and habitats. It is noteworthy that these surveys identify mats of filamentous algae uncharacteristic on the urbanised western side of the island. However, this is a correlational observation and causality cannot be determined. I suggest the authors can provide greater support for the argument that anthropogenic stressors have resulted in this observation by providing greater historical insight into the trend in nature and intensity of anthropogenic stresses since 2000, the point in time that a previous observation is referred to at lines 214-218.

Validity of the findings

The study provides a useful description of rarely studied environments and is therefore novel. However, as a result of the nature of the study, the work is largely descriptive with limited insight to causes of mats of filamentous algae.

Additional comments

I suggest the manuscript can be improved with revisions of the text and encourage the authors to support the conclusions with historical insight to the anthropogenic stressors likely to have resulted in mats of filamentous algae and cyanobacteria.

---

## Round 0.2 · Minor Revisions

Dear Javier and co-authors,

Please revise the remaining minor comments highlighted by two reviewers. Please correct the points made by Reviewer#2 and also make sure the manuscript clearly acknowledges the preliminary/descriptive nature of the study (see Reviewer#3 comment).

I'll be looking forward to receiving your revised manuscript along with a point-by-point response to the reviewers' comments.

With regards,
Xavier

·

Basic reporting

Overall the manuscript has greatly improved. I have a just a couple of small comments on the reporting.
Line 158 - I think the authors can remove the reference to Ford et al. 2021 as the referenced study did not support the authors in identifying their taxa - there are plenty of other manuscripts that assist with identifying cyanobacteria that would be more appropriate to cite here, depending on what the authors used, or the reference could be removed entirely.
Line 198 - please correct spelling of Lyngbya.
Line 223 - I also have one small correction to an earlier comment from myself actually - the correct reference for the added statement on Lynbya sensu lato is Komárek 2016 [Komárek, J. A polyphasic approach for the taxonomy of cyanobacteria: principles and applications. European Journal of Phycology 51, 346–353 (2016)] NOT Duperron et al. 2020 - please ensure to correct this and apologies for this oversight.
Line 259-263 - I find these sentences a bit disjointed and suggest to rearrange (perhaps have all the statement on iron in one sentence, and the shifting N:P ratio is the other).
Line 278 - the study referred to here technically showed the cyanobacteria linked to increased organic matter (not nutrients - though of course the two often are hand in hand). Other references may be more appropriate for linking cyanobacteria to nutrient input.

Experimental design

no comment

Validity of the findings

no comment

Additional comments

Well done on the effort and time spent in improving this manuscript. I think this serves as a good preliminary study from which other work can build upon.

Reviewer 3 ·

Basic reporting

I congratulate the authors on revisions made to the manuscript which has been considerably improved since initial submission.

Experimental design

I continue to hold reservations regarding the nature of the study that remains primarily descriptive as a result of logistical limitations to the experimental design that result from the opportunistic video sampling from vessels.

Validity of the findings

The study provides a basic description of the mesophotic reefs of Rapa Nui and it is useful for this to be made available as a baseline reference. However, the strength of this is limited by the limitations to the experimental design and descriptive nature of the study

Additional comments

This manuscript has been well reviewed since it initial submission and has included a majority of the suggestions made by reviewers as well as revisions to read clearly. The study provides a description of habitats that have previously not been well studied and can provide a baseline reference for future work and I agree with the authors that this is valuable. I also agree that the authors have to some extent included quantitative approaches for the assessment of videos, despite limitation to these. My reservations with recommending publication of the work remain that this is primarily a descriptive study without a well defined experimental design and this limits the conclusions that can be made from the results. Therefore the manuscript may not represent the scientific rigour favoured by PeerJ.

---

## Round 0.3 · accepted · Accept

Dear Javier and co-authors,

I am pleased to accept your revised manuscript for publication in PeerJ. There are a number of inconsistencies in your reference list, which I will let you correct during the proofs process (see annotated pdf).

I would like to thank all reviewers for their time and efforts in improving this manuscript to this stage.

With warm regards,
Xavier